# Phytochemical and Nutraceutical Screening of Ethanol and Ethyl Acetate Phases of Romanian *Galium verum* Herba (*Rubiaceae*)

**DOI:** 10.3390/molecules28237804

**Published:** 2023-11-27

**Authors:** Alexandra-Denisa Semenescu, Elena-Alina Moacă, Andrada Iftode, Cristina-Adriana Dehelean, Diana-Simona Tchiakpe-Antal, Laurian Vlase, Ana-Maria Vlase, Delia Muntean, Raul Chioibaş

**Affiliations:** 1Department of Toxicology, Drug Industry, Management and Legislation, Faculty of Pharmacy, “Victor Babes” University of Medicine and Pharmacy Timisoara, 2nd Eftimie Murgu Square, 300041 Timisoara, Romania; alexandra.scurtu@umft.ro (A.-D.S.); alina.moaca@umft.ro (E.-A.M.); cadehelean@umft.ro (C.-A.D.); 2Research Centre for Pharmaco-Toxicological Evaluation, “Victor Babes” University of Medicine and Pharmacy, 2nd Eftimie Murgu Square, 300041 Timisoara, Romania; 3Department of Pharmaceutical Botany, Faculty of Pharmacy, “Victor Babes” University of Medicine and Pharmacy Timisoara, 2nd Eftimie Murgu Square, 300041 Timisoara, Romania; diana.antal@umft.ro; 4Department of Pharmaceutical Technology and Biopharmaceutics, Faculty of Pharmacy, “Iuliu Hatieganu” University of Medicine and Pharmacy, 8th Victor Babes Street, 400347 Cluj-Napoca, Romania; laurian.vlase@umfcluj.ro; 5Department of Pharmaceutical Botany, Faculty of Pharmacy, “Iuliu Hatieganu” University of Medicine and Pharmacy, 8th Victor Babes Street, 400347 Cluj-Napoca, Romania; gheldiu.ana@umfcluj.ro; 6Department of Microbiology, Faculty of Medicine, “Victor Babes” University of Medicine and Pharmacy Timisoara, 2nd Eftimie Murgu Square, 300041 Timisoara, Romania; muntean.delia@umft.ro; 7Multidisciplinary Research Center on Antimicrobial Resistance, “Victor Babes” University of Medicine and Pharmacy, 2nd Eftimie Murgu Square, 300041 Timisoara, Romania; 8Department of Surgery I, Faculty of Medicine, “Victor Babes” University of Medicine and Pharmacy, 2nd Eftimie Murgu Square, 300041 Timișoara, Romania; office@medcom.ro; 9CBS Medcom Hospital, 12th Popa Sapca Street, 300047 Timisoara, Romania

**Keywords:** antioxidants, DPPH free radical, total phenolic content, FT-IR profile, cytotoxicity, melanoma, HaCaT cells

## Abstract

*Galium* species are used worldwide for their antioxidant, antibacterial, antifungal, and antiparasitic properties. Although this plant has demonstrated its antitumor properties on various types of cancer, its biological activity on cutaneous melanoma has not been established so far. Therefore, the present study was designed to investigate the phytochemical profile of two extracts of *G. verum* L. herba (ethanolic and ethyl acetate) as well as the biological profile (antioxidant, antimicrobial, and antitumor effects) on human skin cancer. The extracts showed similar FT-IR phenolic profiles (high chlorogenic acid, isoquercitrin, quercitrin, and rutin), with high antioxidant capacity (EC_50_ of ethyl acetate phase (0.074 ± 0.01 mg/mL) > ethanol phase (0.136 ± 0.03 mg/mL)). Both extracts showed antimicrobial activity, especially against Gram-positive *Streptococcus pyogenes* and *Staphylococcus aureus* bacilli strains, the ethyl acetate phase being more active. Regarding the in vitro antitumor test, the results revealed a dose-dependent cytotoxic effect against A375 melanoma cell lines, more pronounced in the case of the ethyl acetate phase. In addition, the ethyl acetate phase stimulated the proliferation of human keratinocytes (HaCaT), while this effect was not evident in the case of the ethanolic phase at 24 h post-stimulation. Consequently, *G. verum* l. could be considered a promising phytocompound for the antitumor approach of cutaneous melanoma.

## 1. Introduction

Nowadays, more and more emphasis is placed on the production of herbal medicines for the treatment of human diseases [1,2,3,4] due to old ethnopharmacological knowledge and continuous therapeutic research by scientists. Since ancient times, plants have been used as support in the biomedical domain due to their never-ending pharmaceutical and medical properties, useful in both the prevention and treatment of various diseases. Therefore, plant-based medicinal products continue to attract the attention of researchers around the world due to their beneficial effects on human health as well as minimum side effects in human organisms [5]. According to the World Health Organization (WHO), medicinal plants contribute to obtaining many traditional medicines, which are useful for the primary healthcare needs of millions of people, especially those from developing countries [6,7].

Throughout time, the number of studies involving medicinal plants has increased, emphasizing potential beneficial effects in severe pathologies like cardiovascular diseases, diabetes, pulmonary and brain diseases, and cancer [8,9,10,11,12,13]. Through their mechanism of action in the human body, medicinal plants are responsible for the protective effects due to their capacity to reduce oxidative stress (to reduce intracellular reactive oxygen species (ROS)) and protect cells against the harmful damage caused by H_2_O_2_ [14,15]. Therefore, an intact skin barrier leads to a strong defense against various factors that could create dysfunctionality, thus forming a start for various skin disorders, such as skin cancer [16]. It is well known that skin cancer, especially cutaneous melanoma, has an increased incidence rate globally, being one of the most frequent cancers among young people. Besides genetic factors, it is assumed that environmental ones contribute more to the development of melanoma, like exposure to ultraviolet (UV) radiation from the sun and from tanning lamps and beds [17,18]. In this context, there is an enormous need to find promising alternatives to ensure skin integrity maintenance, including the development of beneficial products for skin disease therapy. Regarding cutaneous melanoma, it is urgent to develop novel anticancer compounds, especially based on plants, thus overcoming the side effects after administration of cytotoxic agents currently used for metastatic melanoma (increased resistance of melanoma cells) [19]. Plant extracts are the perfect candidates for use as alternatives to conventional treatments due to their plethora of natural bioactive compounds and phytochemicals with favorable bioactivity for many human ailments. Plants represent a rich, never-ending natural source of antioxidant compounds, able to counteract oxidative stress and mitigate its effects on individuals’ health. Between antioxidant compounds, polyphenols are considered of major relevance regarding the antioxidant effect for many medicinal plants. The higher the number of phenolic compounds present in an extract, the stronger the supplementary effects of the extract, such as synergistically, additively, or antagonistic actions [20,21,22], that influence the total capacity of the extract to neutralize free radicals [23,24,25].

Among the medicinal plants that have attracted the attention of scientists in recent years, with a long history as a traditional healing plant, is the *Galium* species. *Galium* sp. belongs to the *Rubiaceae* family, representing a safe, accessible, and efficient natural health remedy, proven by its representative bioactive compounds like iridoid glycosides [26], terpenes [27], phenolic acids [28], flavonoids [29], monoterpene glycosides [30], phytosterols, anthraquinones [31], saponins [32], aldehydes, alcohols, small amounts of tannins, waxes, pigments, essential oils [33], and vitamin C [34]. Across the world, there are about 667 species of *Galium*, including in Africa, Asia, North America, and Europe, where over one-third of the species are distributed [35]. In Romania, the genus *Galium* is represented by approximately 38 species, of which 6 have yellow flowers [36,37,38,39]. Among the species from Romania, the best known is *G. verum* L. (lady’s bedstraw), next to *G. mollugo* L. (hedge bedstraw), *G. aparine* L. (cleavers or stickyweed), and *G. odoratum* L. (syn. Asperula odorata, woodruff) [40]. Due to the beneficial effects of *Galium* spp. reported over time, nutraceuticals (galenic remedies and supplements) are nowadays available on the market that are supposed to contribute to improving some health issues involving inflammation, detoxication, or the immune system. The most well-known pharmacological activities of *Galium* species have been reported in a previous study and refer to the biological effect of *Galium verum* L. extract internal and external administration [40]. In traditional medicine, *G. verum* L. has been used for its depurative, diuretic (for bladder and kidney irritation), laxative, antirheumatic, and sedative actions [26,38], as well as for healing wounds and gingival inflammations [41], epilepsy and hysteria [42]. In addition, in Romanian traditional medicine, nutraceuticals from several *Galium* spp. have been used for different cosmetic formulations to prevent the actions of oxidative stress on the skin [40]. Several scientific studies have reported that the representatives of the *Galium* genus possess a wide range of biological properties, including antioxidant, detoxicant, antimicrobial, antifungal, antihaemolytic, cardio- and hepatoprotective, immunomodulatory, and anticancer properties [43,44,45,46,47,48,49,50,51,52]. In addition, other scientific studies reported that *Galium verum* L. extracts have been used to treat skin disorders, as exogenous treatment in psoriasis, and as a treatment in the healing of delayed diseases. Moreover, it was shown that these species have beneficial effects on cancer ulcers, tongue cancer, and breast cancer. Currently, the extract of *Galium verum* L. is recommended for therapy in rheumatic diseases and cystitis [27,44,53]. To the best of our knowledge, no study in the scientific literature reports the in vitro antitumor effect of *Galium verum* L. extracts on cutaneous melanoma. Based on the findings reported previously and considering the lack of information, the present study aims to report for the first time the preliminary results regarding the in vitro antitumor effect of *G. verum* L. extracts on skin disorder, more precisely, the in vitro biological effect on malignant melanoma.

In this context, the current study aimed to outline the phytochemical profile of two extracts obtained from the aerial part of *Galium verum* L. plant material acquired from a local natural products store, including the individual phenolic compounds content and the antioxidant screening, together with a preliminary biological evaluation regarding the antimicrobial activity and the in vitro anticancer effect. The novelty of the study consists in the fact that, although this plant species has shown great antitumor potential on various cancer cell lines, its efficacy in skin cancer has not yet been established. Therefore, one of the objectives of the present study was to assess the anticancer effect of *Galium verum* L. on the human skin cancer cell line A375, as well as on the healthy human keratinocyte cell line (HaCaT).

## 2. Results

### 2.1. Phytochemical Analysis

After the concentration of both *Galium verum* L. extracts, the extraction yield was calculated, followed by Fourier transform infrared spectroscopy (FT-IR) characterization. In addition, the phenolic composition and antioxidant potential to establish the phytochemical profile of the obtained extracts were determined.

#### 2.1.1. The Extraction Yield

The extraction was carried out using a conventional extraction procedure based on maceration of *Galium verum* L. plant material for 24 h at room temperature, followed by sonication (at 25 °C for 30 min), then filtration and concentration of a specific volume of each extracted sample. The extraction yield calculated from 50 mL of each *Galium verum* L. extract was 13.95% for GvEtOH extract and 4.65% for GvEtOAc extract, respectively. By analyzing the values obtained regarding the two extracts, it is clear that the final extraction yield depends on the type of extraction solvent, the plant:solvent ratio, and the amount of vegetal material taken into account. It was chosen to work with dried vegetal material instead of the fresh one because the dried plant has a lower amount of water as compared to the fresh one, water which is lost together with volatile compounds in the drying step.

#### 2.1.2. FT-IR Investigations

The molecular fingerprint of the *Galium verum* L. extracts resulting from the signal recorded by each molecule or chemical structure at a specific wavenumber is depicted in Figure 1.

The results of the FT-IR analysis for both concentrated *Galium verum* L. ethanol extract and ethyl acetate phase are provided in Table 1.

The FT-IR analysis recorded strong absorption bands, especially in the case of GvEtOH extract, at around 3400, 1700, and 1070 cm^−1^. The first important, strong, and well-defined band is located at 3412.08 cm^−1^ and can be attributed to the O-H stretching vibration (hydroxyl groups (H-bonded)) present in water or flavones contained in the ethanolic extract of concentrated *Galium verum* L. plant material. The bands located around 2920 cm^−1^ in both extracts can be assigned to the O-H stretching in acid functional groups or to saturated aliphatic C-H stretching bonds (bands around 2852.72 cm^−1^), suggesting the occurrence of aromatic ring attachment. The bands around 1730 cm^−1^ present in both extracts indicate the presence of carbonyl functional groups, and the band recorded at 1654.92 cm^−1^ from the GvEtOH extract indicates that the C=C group is present in the alkenes, but it could also be assigned to the C=O stretching vibration of amide functional groups. The band recorded at 1604.77 cm^−1^ in both extracts suggests the presence of C=C stretching vibration of cyclic alkenes. The medium–weak intensity absorption peaks recorded between 1300 and 1600 cm^−1^ revealed the presence of the following functional groups: C=C stretch (in a ring) and C-H bending from aromatic compounds and alkanes (methylene group). The bands recorded around 1260 cm^−1^ are assigned to the stretching vibration of the C-O functional group, most probably from the aromatic carbonyl acids. Bands between 1070 and 1170 cm^−1^ are assigned to primary, secondary, and tertiary alcohols, and the bands recorded between 800 and 1045 cm^−1^ are attributed to the C=C bending, C-H bending, and CO-O-CO stretching vibration present in alkanes (di- and trisubstituted/alkenes and anhydrides functional groups. The region between 550 and 810 cm^−1^ is specific for the out-of-plane stretching vibration of halo compounds (C-Cl and C-Br stretching), as well as to the bending vibration of C=C functional groups present in the alkane aromatic compounds.

#### 2.1.3. Liquid Chromatography Mass Spectrometry (LC-MS)

Table 2 presents the polyphenolic content of the *Galium verum* L. extracts (ethanolic and ethyl acetate phases) obtained by LC-MS analysis.

LC-MS analysis revealed seven phenolic compounds in both extracts of *Galium verum* L., identified as major components in the analyzed extracts. Quercitrin was the only phenolic compound that was below the limit of quantification in the GvEtOH extract. Obtained results indicated that isoquercitrin and rutin (two important flavonoids) were the most abundant quantified compounds in GvEtOH extract compared with chlorogenic acid and isoquercitrin (a phenolic acid and a flavonoid), which were the most abundant quantified compounds in GvEtOAc extract. Therefore, the GvEtOAc extract was richer in terms of phenolic compounds than GvEtOH, except 4-O caffeoylquinic acid, rutin, quercetol, and luteolin. In both extracts, one can observe that the amount of flavonoids is higher than that of phenolic acids, quantified in Table 2.

The chemical structures of the polyphenols found in both extracts were designed with the KingDraw Chemical Structure Editor (http://www.kingdraw.cn/en/ accessed on 16 November 2023) and are presented in Figure 2. One can also observe the organic functional groups identified with FT-IR analysis.

Table 3 reveals the results for the identification and quantifications of catechins from both *Galium verum* L. extracts, analyzed by LC-MS. One can observe that epicatechin and gallic acid were the only polyphenolic compounds detected in *Galium verum* L. extracts in low concentrations. The rest of the polyphenolic compounds were below the detection limit.

#### 2.1.4. Total Phenolic (TPC) and Flavonoid Contents (TFC)

Depending on the solvent used for the extraction of polyphenolic compounds from *Galium verum* L. herba, the content of the total phenols slightly varied. The TPC in GvEtOH extract was 1.30 mg GAE/g dry extract as compared with the TPC in GvEtOAc extract of 1.39 mg GAE/g dry extract. Regarding the total flavonoid content, the same slight variation was observed. The TFC in GvEtOH extract was 1.42 mg CE/g dry extract as compared with TFC in GvEtOAc of 1.37 mg CE/g dry extract.

#### 2.1.5. Antioxidant Activity

Figure 3 depicts the degradation kinetics of DPPH free radicals provided by both *Galium verum* L. extracts, evaluated each at six different concentrations, as well as by the standard used: the ethanolic solution of ascorbic acid (vitamin C). One can observe that the degraded amount of DPPH free radicals after 20 min of incubation period indicated a 12% degradation when the GvEtOH extract was used at the highest concentration tested (1 mg/mL), while at the same concentration, the GvEtOAc showed 4% degradation of DPPH free radicals, a degradation potential almost identical with the one observed when vitamin C was used (3%). The same degradation potential of DPPH free radicals was observed when using the GvEtOAc extract at a concentration of 0.8 mg/mL, even at 0.5 mg/mL (7%), which indicates that, by using ethyl acetate as an extraction solvent, more antioxidant compounds are extracted from the dried plant, even at low concentrations. Nevertheless, both assays showed a similar trend, with GvEtOAc extract presenting the highest antioxidant potential. In addition, in the case of both extracts, the DPPH free radicals are consumed in the first 300 s (for the extracts with high concentration, between 0.5 and 1 mg/mL), and then the kinetics of the reaction reach equilibrium. In the case of the last three concentrations (0.05–0.3 mg/mL), the DPPH free radicals are consumed faster, reinforcing the assumption that the less concentrated extracts contain a reduced amount of antioxidants and phenolic compounds.

The antioxidant potential of both *Galium verum* L. extracts, compared with the antioxidant potential of ascorbic acid ethanolic solution (vitamin C), is shown in Table 4. The antioxidant potential percentage obtained for all the concentrations tested of *Galium verum* L. extracts (ethanol and ethyl acetate fraction) represents an average of three measurements ± standard deviation (SD). Further, by linear regression analysis, between these values and their concentrations, the EC_50_ was calculated. The EC_50_ of *Galium verum* L. ethanol extract was 0.136 ± 0.03 mg/mL (R^2^ = 0.92046), and the EC_50_ of ethyl acetate fraction was 0.074 ± 0.01 mg/mL (R^2^ = 0.94174).

One can observe that both *Galium verum* L. extracts show antioxidant potential well above 35% in the case of GvEtOH samples, whose values were lower than in the case of the samples obtained from the ethyl acetate fraction (GvEtOAc), which starts at above 45% at the smallest concentration tested (0.05 mg/mL). Moreover, it can be observed that in the case of GvEtOAc extract, the samples of 1 mg/mL and 0.8 mg/mL revealed values (96%) almost identical to the standard value of Vit C (97%) at a concentration of 0.4 mg/mL.

Considering the results obtained, it can be noticed that the antioxidant potential of all the samples tested, obtained from both *Galium verum* L. extracts, are concentration dependent.

### 2.2. Bioactivity

#### 2.2.1. Antimicrobial Analysis

The obtained results for the antibacterial activity of *Galium verum* L. plant extracts against Gram-positive and Gram-negative bacilli strains are presented in Table 5. The antimicrobial effect was measured by micro-dilution assay, and the MIC (mg/mL) and MBC (mg/mL) determination were assessed. The MIC values obtained for the *Galium verum* L. ethanol extract ranged from 15 to 30 mg/mL, as well as for the *Galium verum* L. ethyl acetate extract.

The results obtained revealed that both extracts have only a bacteriostatic effect on the Gram-negative *Escherichia coli* strain and no antimicrobial activity on the Gram-negative *Pseudomonas aeruginosa* strain. By comparing the two extracts, it seems that the GvEtOAc extract showed better antimicrobial activity against the Gram-positive bacilli strains used.

#### 2.2.2. Anticancer Potential

##### Viability Assay

To analyze the ability of the extracts to inhibit cell proliferation, the MTT assay was performed. In all cases, the viability percentages varied in a concentration-dependent manner. The effect of *Galium verum* L. extracts (five concentrations from the two phases) ethanol and ethyl acetate, the lowest concentration used was 15 μg/mL, followed by 25 μg/mL, 35 μg/mL, and 55 μg/mL was the highest concentration used) was evaluated on a healthy cell line (skin immortalized keratinocytes—HaCaT) and the human malignant melanoma cell line A375 and compared with the control group, untreated cells.

Figure 4 shows the effect of GvEtOH and GvEtOAc extracts on HaCaT cells after a 24 h stimulation period. It can be seen that at the lowest tested concentrations (15 and 25 μg/mL), the ethyl acetate phase from *G. verum* L. produced a significant increase in cellular viability compared to the control, more precisely 121%, and 103.9%, respectively, but concerning the ethanolic extract, no increase was observed but a slight decrease in cell viability of 97.4% and 92.6%; from these data, it can be stated that GvEtOAc can stimulate the proliferation of a healthy cell line.

Further, Figure 5 illustrates the effect of the extracts on A375 melanoma cancer cells after 24 h of stimulation. Treatment with the two extracts caused a dose-dependent decrease in tumor cell viability. At the first concentration, the GvEtOH extract shows a minor and insignificant increase in cell viability at 100.8%, followed by 95.4%, while at the same doses, the ethyl acetate phase gradually decreases cell viability at 90.3% and 86.2%. Additionally, at the highest concentration tested, tumor cell viability decreased compared to the control to 60.3% for GvEtOAc, while the GvEtOH extract showed a decrease in viability of 77.8%.

The results obtained indicate that the GvEtOAc extract affects more the skin tumor cells than the healthy cell line, while in the case of the ethanolic phase, there were less obvious differences between the effect observed on tumor and non-tumor lines following the MTT assay.

##### Cell Morphology and Confluence

As a component of the antitumor profile of GvEtOH and GvEtOAc extracts, a microscopic examination of HaCaT (Figure 6) and A375 cells (Figure 7) was performed after 24 h of treatment.

Because no significant changes in viability were observed between the middle concentrations tested, we decided to further evaluate and highlight the morphological aspect at three of the most suggestive concentrations (15, 35, and 55 μg/mL). A bright field microscope analysis of confluence and cell morphology was performed after 24 h of stimulation to provide an overview of the effects of the two extracts. On healthy skin cells, GvEtOH and GvEtOAc produced no significant changes in cell morphology, with cells remaining similar to untreated control cells, and had no negative impact on cells’ adherence or confluence. For GvEtOAc, at the lowest dose, an increase in cell confluence was observed, while at concentrations of 35 and 55 µg/mL, a slight decrease in confluence was seen. Referring to the GvEtOH extract, a dose-dependent decrease in cell confluence was observed, with minor cell damage at the concentration of 55 µg/mL, data that are consistent with the cell viability results.

Instead, in the case of tumor cells, changes in shape and confluence were visible, depending on the extract and the tested concentration. For the GvEtOH extract, the highest concentration decreased cell confluence with changes in shape. While cells treated with 35 µg/mL and more with 55 µg/mL of the GvEtOAc extract visibly lost their shape and became round, several signs of cell death were observed, with the cells detaching from the plate and decreasing the confluence.

Therefore, differences were observed between treated and untreated cancer cells depending on the tested dose; the concentration of 55 µg/mL had a direct impact on the cell morphology and the number of cells.

##### Nuclear Morphology Evaluation

The last step in our study was to evaluate the cell death mechanisms underlying the cytotoxic effect of the tested extracts by examining the appearance of the nuclei of HaCaT and A375 cells using Hoechst 33342 dye.

Starting from the fact that the highest concentration tested caused a reduction in cell viability, we continued to evaluate the *Galium verum* extracts to identify whether cell death occurred by apoptosis or necrosis, comparing the effect produced by the first and the last dose of the two phases. This analysis was carried out to outline in more depth the activity of the GvEtOH and GvEtOAc extracts on HaCaT and A375.

Neither extract visibly affected the healthy skin line at the level of the nuclei. No apoptotic bodies, cell shrinkage, or nuclear fragmentation were evident, even at the dose of 55 µg/mL (Figure 8).

At the tumor cell level, the extracts showed damage to the nuclei at the dose of 55 µg/mL. In the case of the control cells, the nuclei have a regular shape, uniformly colored, without signs of fragmentation, this aspect being exposed even at the lowest concentration of the extracts. For the GvEtOAc phase, it was observed that at the highest concentration, signs of cell apoptosis were present, with nuclear condensation and the appearance of apoptotic bodies. In contrast, the GvEtOH extract induced less visible dysmorphology and did not cause as significant of changes in the nuclei. Chromatin condensation was recorded at the concentration of 55 µg/mL (Figure 9), and changes were observed after performing the Hoechst test.

Yellow arrows indicate the results regarding apoptotic characters. The results of nuclear morphology assessment are also expressed as apoptotic index (AI). A concentration of 55 µg/mL induced an increase in the percentage of apoptotic index in the A375 cell line compared to the control, where no signs of apoptosis were detected. For the GvEtOH and GvEtOAc extracts, the percentages recorded by AI were 14.17% and 29.67%, respectively.

## 3. Discussion

Worldwide, most people emphasize the use of herbs for the treatment of any health condition, according to the World Health Organization [54]. Therefore, the investigation of the pharmacological effects of vegetal material represents a continuous concern for many researchers. Vegetal materials are sources full of natural phytocompounds with a plethora of beneficial activities for the living organism, such as antimicrobial, anti-inflammatory, antioxidant, and/or anticancer properties, bioactive substances that could contribute to the manufacturing of efficient and safe therapeutic drugs [55]. Polyphenols are the most important class of natural bioactive phytocompound and are considered the secondary metabolites that are either normally synthesized by plants during their development or are by-products that occur as a response to the ecological stress factors to which the plant is subjected (like pollution) [56]. Besides polyphenols, flavonoids, phenolic acids, anthocyanins, tannins, terpenes, saponins, vitamins, or essential oils are natural antioxidants with an important role in protecting biological systems against the harmful consequences of oxidative stress. The plant species from the Rubiaceae family, genus *Galium*, represent a valuable source of polyphenolic compounds [38]. In this context, the present study was conducted to evaluate the phytochemical and biological profile of two phases of *Galium verum* L. plant material acquired from a local store by assessing their phenolic content and their antioxidant potential correlated with their possible relevant bioactivities, like antimicrobial and in vitro anticancer effects.

It is well known that extraction techniques represent a key factor in collecting as many polyphenols as possible from a vegetal material. Therefore, the extraction technique is mainly dependent on the quality of the vegetal material, the solvent used in the process, the extraction procedures chosen, and the equipment used. These critical parameters will define the quality of the extract and the extraction yield. To achieve a high extraction yield, the type of solvent, the plant particle size, the temperature, and the duration of the extraction process are factors that can be modulated. Regarding the best extraction technique used for collecting a high amount of polyphenols from plant material, the conventional methods based on solid–liquid extraction with different solvents are still the most desired and used techniques by researchers due to their easy-to-use, efficient, and wide-ranging applicability for any vegetal material [57]. Therefore, the present study employs the classical solid–liquid extraction method, based on maceration, sonication, filtration, and concentration of the final product, using both the ethanol 95% and concentrated ethyl acetate as solvents. To obtain a qualitative extract without traces of heavy metals or toxic compounds due to the soil components or soil contaminants, the plant material was acquired from a local store. Based on the results obtained after the extraction yield for both phases was calculated (13.95% for GvEtOH extract vs. 4.65% for GvEtOAc extract values obtained from 50 mL extract), one can say that the extraction yield depends mainly on the type of solvent used. Besides solvent type, in our case, even the amount of the plant material (25 g for GvEtOH vs. 10 g for GvEtOAc), as well as the plant:solvent ratio, were defining factors for the yield and the rate of polyphenols obtained [58,59,60]. Even if it is known that methanol is more efficient for low-molecular-weight polyphenols extraction and acetone for high-molecular-weight polyphenols extraction, other solvents like ethanol, ethanol–water, ethyl acetate, etc., can be considered [57,61] because they are volatile solvents, safe and friendly with the environment, and often used in pharmaceutical applications. Our results regarding the extraction yields are higher than the results reported in the literature, most probably due to the quality of the plant material [34]. The group conducted by Lakić found that the methanolic extract of *Galium verum* L. wild growing in two different cities of Serbia had extraction yield ranging from 5.44 to 7.21% (g of dried extract/100 g sample), depending on the duration of extraction [34]. A result similar but slightly higher compared with ours (13.95% in the case of GvEtOH extract) was obtained by Friščić and co-workers [62]. The differences come from the type of solvent used; the authors used 80% methanol and obtained an extraction yield of 18% by employing the ultrasound-assisted extraction technique after a long period of plant material maceration.

The investigation methods play an important role in the establishment of a basic, detailed profile regarding the beneficial therapeutic activity of a plant extract and explain to what extent one or more compounds contribute to the appearance of the biological effect. Generally, the biological activity of a plant extract is correlated to the concentration of phytocompounds. The phytochemical analyses contribute to the detection of plant compounds, thus preventing their use and improper manipulation from the point of view of the biological effect. Therefore, in the present study, we performed the analytical technique of liquid chromatography coupled with mass spectrometry (LC-MS), which separates, identifies, and quantifies the polyphenols (phenolic acids and flavonoids) from *Galium verum* extracts. In addition, to identify the unknown organic, polymeric, and, in some cases, inorganic compounds or additives and contaminants, we employed another analytical technique: Fourier transform infrared spectroscopy.

The polyphenolic profile of the two phases was analyzed using liquid chromatography coupled with mass spectrometry detection (LC-MS). The LC-MS is an analytical method used for the identification of different classes of polyphenolic compounds by a single pass of the plant extract on a reverse-phase analytical column in a short time [63]. The outcomes obtained for both phases of *Galium verum* L. plant material highlighted the identification of seven polyphenolic compounds in both *Galium verum* L. extracts, followed by their quantification based on their peak area and the calibration curve of their corresponding standards. Both extracts have shown identical qualitative phenolic profiles regardless of the type of solvent used for extraction, the only differences being quantitative. Rutin was found in higher quantity in GvEtOH extract (14.811 μg/mL) compared to GvEtOAc extract (1.896 μg/mL). Chlorogenic acid and isoquercitrin were more abundant in GvEtOAc extract (10.216 μg/mL and 20.384 μg/mL) as compared with GvEtOH extract (8.027 μg/mL and 17.765 μg/mL). The rest of the polyphenols and flavonoid compounds were found in low concentrations or below the limit of detection (e.g., quercitrin in GvEtOH extract as compared with 6.722 μg/mL quercitrin found in GvEtOAc extract). It was stated that rutin, an important flavonoid glycoside, has effectiveness in various diseases such as inflammatory bowel disease, Alzheimer’s disease, and cancer [64,65,66], while isoquercitrin (a flavonoid glycoside) possesses suitable well-known anti-inflammatory effects [67]. Our findings are in agreement with those reported in previous studies [38,68,69].

As a completion of the polyphenolic profile, the functional group’s fingerprints were investigated by FT-IR. This analysis can be seen as a plus because the chemical structure of the phenolic compounds is related to the antioxidant capacity of the extracts, especially the number of available hydroxyl groups as well as the extract concentration. Although both phenolic acids and flavonoids are known strong antioxidant compounds, their ability depends on the chemical structure more than the amount in which they are present in a sample [70]. Therefore, the most important functional groups were recorded at 3412.08 cm^−1^, which are assigned to the hydroxyl groups from alcohols, phenols, and carboxylic acids [71]. This band usually signals the presence of flavones, such as rutin. The double band recorded at 2926.01/2852.72 cm^−1^ indicates the occurrence of the aromatic ring and alkyl group attachment to the C-H stretching functional group or indicates the O-H functional groups from phenolic acids, such as chlorogenic and/or 4-O caffeoylquinic acids. However, at the same time, the band situated at 2926.01 cm^−1^ may correspond to the CH_3_ vibrations, which exist in the functional groups of chlorophyll present in the *Galium verum* L. plant material [72]. The peaks recorded at 1516.05 cm^−1^ confirm the presence of the aromatic ring in both extracts [73]. The bands situated at 1259.52 cm^−1^ and 1261.45 cm^−1^ correspond to the aromatic acid ester C-O stretching vibration [74]. These bands are attributed to flavones like rutin or even to chlorogenic acid since it is the ester of caffeic acid because flavonoids/flavones contain ester-like functional groups in their structure. The primary, secondary, and tertiary alcohol functional groups from both extracts are highlighted by the peaks recorded between 1070 and 1170 cm^−1^ [74]. The bands recorded between 550 and 810 cm^−1^ could be the out-of-plane bending vibration from aromatics compounds [75], which in this case could be specific for aromatic bicyclic monoterpenes or of halo compounds (C-Cl and C-Br stretching). The presence of halo compounds may be due to the presence of some soil minerals that have been absorbed by the plant or even to some impurities from the glassware used during extraction, respectively, during characterization.

The total content of phenols and flavonoids was also investigated. The total phenolic content method is based on the electron transfer reactions between the Folin–Ciocalteu reagent and phenolic compounds present in both *Galium verum* L. extracts when a blue-colored complex is formed and can be quantified spectrophotometrically [76,77]. Our results regarding the total phenolic and flavonoid contents showed a small content of polyphenolic compounds but agreed with the results obtained from LC-MS analysis. We have obtained a TPC of 1.30 mg GAE/g dry extract for the GvEtOH extract as compared to 1.39 mg GAE/g dry extract for the GvEtOAc extract. In the case of TFC quantification, 1.42 mg CE/g dry extract was obtained for the GvEtOH extract, compared to 1.37 mg CE/g dry extract for the GvEtOAc extract. It seems that our results are similar to the results obtained by Danila and co-workers [78] but much lower than the results of other researchers reported in the scientific literature [38,62,68,79,80]. Similar results but slightly higher regarding the TPC were reported by Mocan and co-workers [38], although the authors found an increased amount of chlorogenic acid as well as rutin through LC-MS analysis. Also, similar but higher results regarding the TPC were reported by Lakic and co-workers [34] and Vlase and co-workers [81]. Regarding the TFC, Laanet and co-workers obtained 2.6 mg QE/g extract for an extract of 80% ethanol of *Galium verum* L. herb. These differences may be due to the extraction method applied (duration, temperature), the concentration and type of solvent used, the geographical area and soil content where the plant has grown (quality of plant material), the amount of the plant material used in the extraction process, and many other factors that can influence the total phenolic and flavonoid contents.

It is known that plants are the most abundant and cheapest source of food and medicinal cures for humans. For medicinal purposes, the plants are used as a source of antioxidant compounds, like polyphenols, phenolic acids, and/or flavonoids, capable of scavenging the free radicals by inhibiting the oxidative stress that leads to a variety of human diseases (asthma, diabetes, Alzheimer’s and Parkinson’s diseases, atherosclerosis, inflammatory arthropathies, and cancers). According to Armatu et al. [82], a plant material that has a low content of antioxidant compounds proves the fact that the plant has a weak polyphenols content and vice versa. Therefore, a strong antioxidant capacity correlated with a high content of polyphenols leads to a high potential of plant material to prevent oxidative stress and to confer excellent anti-inflammatory, antibacterial, and/or antitumor properties. There are various methods to assess the antioxidant potential of *Galium verum* L., reported either as IC_50_, original absorbance or % loss, or Trolox/ascorbic acid equivalent [38,40,68,80]. Regardless of the comparison method (direct—through a concentration-dependent expression or indirect—through comparison with a standard), through the antioxidant compounds assessment, one can determine the health benefits of a plant material as well as the promising future properties for the curing disease, regarding the people who consume it. For a simple and fast estimation of the antioxidant potential, it was employed the DPPH free radical-scavenging assay (2,2-diphenyl-1-picrylhydrazyl), an in vitro non-cellular assay, due to its wide use, as well as stability, reliability, and the simplicity of the assay [83]. It has been showed that the assessed extracts obtained from *Galium verum* L. herba, acquired from a local natural products store, were able to reduce the purple stable radical DPPH· to the yellow colored DPPH-H, reaching 50% reduction with an EC_50_ of 0.136 mg/mL in the case of GvEtOH and 0.074 mg/mL in the case of GvEtOAc, respectively. Previous studies of the antioxidant potential of *Galium verum* L. extracts showed a similar trend, with lower IC_50_ values established in the DPPH test. For instance, Lakić and co-workers found that the methanolic extracts obtained at different hours of extraction showed an IC_50_ ranging from 3.10 μg/mL to 8.04 μg/mL, depending on the geographical area from where the lady’s bedstraw was collected [34]. In addition, Friščić and co-workers reported an IC_50_ of 30.72 μg/mL for *Galium verum* L. extract, 80% in methanol [62]. A similar trend (with lower IC50 values) has also been reported in previous studies regarding the antioxidant activities of the extracts obtained from *Galium* species established by the DPPH test [84,85]. These differences observed between the values regarding the DPPH free radical test could be the result of either the concentration of DPPH used, its greater sensitivity to reaction conditions, the geographical area, the age of the plant, the date of collection, the soil where the plant was grown, the type and volume of solvent used for compounds extraction/isolation, and the type and parameters of extraction method employed or the plant material:solvent ratio used. All these factors may contribute to the obtaining of a high inhibitory concentration value of the plant’s antioxidants, which are needed to scavenge DPPH free radicals. Moreover, the antiradical capacity of *Galium* species is highly dependent on their flavonoid levels. Rutin (quercetin 3-O-rutinoside), which turned out to be the second predominant flavonoid from *Galium verum* L. ethanolic extract (GvEtOH), might have contributed to the observed antioxidant activity (Figure 3A, Table 4), comparable with the antioxidant activity of GvEtOAc extract [81,86]. Almost identical results with ours (0.136 mg/mL) are reported by Vlase and co-workers for the DPPH free radical-scavenging activity, recorded for the 70% (*v*/*v*) ethanolic extract of *G. verum* L. (105.43 μg/mL) [81]. These results may be the consequence of isoquercitrin (quercetin 3-O glucoside) content instead of quercitrin (quercetin-3-O-rhamnoside), which was recorded below the detection limit by LC-MS analysis, in the case of ethanolic extract. This affirmation contradicts the statements made by Li and co-workers [87], which affirmed that quercitrin exhibited greater activity than isoquercitrin in an H-donating-based 1,1-diphenyl-2-picrylhydrazyl radical-scavenging assay. In addition, the results obtained in the case of the ethyl acetate phase support and strengthen the statement made because, in the case of GvEtOAc extract, the content of isoquercitrin (20.384 μg/mL) was much higher than the content of quercitrin (6.722 μg/mL). A possible explanation could be the difference between the two forms of DPPH (DPPH-I (m.p. 106 °C) is orthorhombic as compared with the DPPH-II (m.p. 137 °C), which is amorphous)). Based on the results obtained, one may say that the phenolic compounds contributed significantly to the antioxidant capacity of the *Galium verum* L. medicinal herbs due to their high potential to neutralize free radicals. This statement is also confirmed by Lakić and co-workers [34], who have investigated the amount of total chlorophylls (a + b) from *G. verum* L. 80% methanol extract, finding the higher amount of total chlorophylls, which partially explains the stronger effects of the plant material on the neutralization of DPPH.

The antimicrobial investigation of the two extracts was performed because, among the phytochemicals of plant material, polyphenols are the most potent antimicrobial compounds, especially phenolic acids and flavonoids. Because both extracts of *Galium verum* L. presented considerable amounts of polyphenols, the antimicrobial activity was performed through the disc-diffusion assay. The disc-diffusion assay was carried out against a panel of microorganisms, including two Gram-positive bacteria, *Staphylococcus aureus* and *Streptococcus pyogenes*, and two Gram-negative bacteria, *Escherichia coli* and *Pseudomonas aeruginosa*. In this regard, we investigated the antimicrobial activity against the aforementioned strains involved in skin pathology. Our findings showed that both *Galium verum* L. extracts were effective against both the Gram-positive and Gram-negative bacilli strains used, especially on *Streptococcus pyogenes* (Gram+). On *Escherichia coli* and *Pseudomonas aeruginosa* strains, both extracts have no antimicrobial activity or only bacteriostatic activity, probably because the Gram-negative strains are more resistant due to the more complex structure of the bacterial wall. The *Pseudomonas aeruginosa* strain, against which it did not obtain an antibacterial effect, is a natural strain resistant to numerous antibiotics, such as narrow-spectrum cephalosporins, ceftriaxone/cefotaxime, tetracyclines, and trimethoprim-sulfamethoxazole. Our findings are contradictory to the data described in the literature. For instance, Shynkovenko and co-workers [32] stated that the *Galium verum* L. 96% ethanol exhibited high antimicrobial activity in the *Pseudomonas aeruginosa* Gram-negative strain as compared with our results, where it was showed that the 95% ethanolic extract of *Galium verum* L. has no antimicrobial activity on the as mentioned bacilli strain. The contradictory results are probably due to other factors, which might play an essential role in the extraction process. However, Vlase and co-workers [81] have reported that the nature of the solvent might play an important role in the antimicrobial activity of ethanolic extract (70%) of *Galiumn verum* L. In addition, Ilyina and co-workers [86] complete Vlase’s affirmation by reporting a significant level of antimicrobial properties of chloroform extract of *G. verum* L. However, in any case, one cannot say with certainty that a *G. verum* L. extract is more or less efficient in Gram-positive or Gram-negative bacilli strains. However, previous studies reported that the Gram-positive bacteria were more sensitive in contact with plant extract, a fact also observed in the present study, probably due to the single layer of the cell walls of Gram-positive bacteria [88,89].

Cultured cell lines represent important biological models for the primary in vitro screening of nutraceuticals that are supposed to be used as possible anticancer candidates. The cultured cell lines can be seen as platforms used in preclinical studies to provide a similar situation suitable for clinical studies. The in vitro methods allow us to investigate the biological potency of the tested phytocompounds from plant material, as well as their mechanism of action within the human organism. Phenolic compounds have an important role in the protection mechanism against pathogens and ultraviolet radiation. Human skin is exposed to ultraviolet radiation, thus increasing ROS (reactive oxygen species) generation, excessive release, and inflammation in cells, which are involved in the pathogenesis of multiple skin disorders like skin cancer. Therefore, many research studies are focused on the investigation of antioxidant compounds that are known to attenuate the damaging effects of ROS. There is a wide variety of in vitro models used as screening studies and mechanistic investigations, but the in vitro assays with human malignant melanoma cells employed in the present study were chosen to offer preliminary valuable information regarding the cytotoxicity of the two extracts of *Galium verum* L. because this plant has not been investigated so far on skin cancer. In addition, another in vitro assay applied in the current study referred to programmed cell death, namely apoptosis, with nuclear fragmentation and formation of apoptotic bodies. These in vitro techniques can only estimate the safety and anticancer activity of the tested extracts based on *Galium verum* L. plant material.

According to the scientific literature, it can be observed that the extracts obtained from *Galium verum* L. have been studied for their antitumor effect on several cancer cell lines, but the effectiveness of the plant in skin cancer is not being scored. Melanoma is a malignant tumor of the skin and has a poor prognosis, mainly due to its high resistance to therapeutic agents. For this reason, finding new antitumor compounds is necessary to improve the evolution of the pathology, which represents an international health problem [90].

The in vitro tests performed in the present study demonstrate that *Galium verum* L. extracts suggest a potential dose-dependent cytotoxic effect against A375 cells, a more pronounced activity being recorded for the GvEtOAc phase, while the GvEtOH extract decreased viability to a lesser extent but considerable. Moreover, the ethyl acetate phase stimulated the proliferation of human keratinocytes, while this effect was not evident in the case of the ethanolic phase at 24 h post-stimulation. An important aspect is the data obtained for non-tumor cells, where the extracts had no significant cytotoxic effect 24 h post-stimulation. Furthermore, knowing that the modification and alteration of cell morphology are signs that attest to cytotoxicity, in the present study, we exposed this type of changes due to *G. verum* L. extracts, especially at the concentration of 55 µg/mL. In addition, we revealed specific characteristics of apoptosis with the help of the Hoechst 33342 method, through which nuclear changes from condensation, cleavage, and the formation of apoptotic bodies can be highlighted [91].

Our results regarding the bioactivity of the plant *G. verum* L. on cancer cells are consistent with the data found in the literature. For instance, Schmidt et al. [53] studied the in vitro effect of *G. verum* L. aqueous extract on chemosensitive laryngeal carcinoma cell lines (Hep-2 and HLaC79) and on chemoresistant laryngeal carcinoma cell lines with P-glycoprotein overexpression (Hep2-Tax, HLaC79-Tax), where they highlighted an effect of inhibiting the growth of both types of cell lines. The extract significantly inhibited the invasion of Hep-2 and Hep2-Tax cells in the collagen gels and extracellular matrix substrates tested, the strongest effect being in the Hep2-Tax cell line. On the other hand, the extract did not affect endothelial tube formation. Furthermore, in this study, gene expression profiling did not reveal unique patterns of gene stimulation or suppression in HLaC79 and Hep2 cell lines [53]. Another study reported by Pashapour and his team highlighted the antitumor potential of *Galium verum* L. extracts on colon (HT29) and liver (HepG2) cancer cell lines. More precisely, it was shown that the fractionated extract with chloroform showed cytotoxic effects on HT29 at the highest concentrations (50 and 100 µg/mL) after a 72 h stimulation; instead, it increased the viability of HepG2 cancer cells. Regarding the petroleum ether fraction, it was observed that on the colon cancer line, it possessed cytotoxic action at all tested doses and on liver tumor cells at the lowest concentration of 3.125 µg/mL. According to the data presented by the researchers, the fractionated extracts exert cytotoxic action against the HT29 and HepG2 cell lines [92]. The group led by Shinkovenko emphasized the immunomodulatory activity of three ethanolic extracts (ethanol 20%, 60%, and 96%) of lady’s bedstraw. All extracts had a stimulating action on the transformation activity of immunocompetent blood cells. The strongest immunomodulatory effect was established for the concentrated extract (96%). Under the influence of this extract, the percentage of blast transformation of lymphocytes increased 6.77–8.04 times compared to their spontaneous transformation and 1.14–1.36 times compared to phytohemagglutinin [47]. Moreover, the same types of ethanolic extracts but from the species *G. aparine* L. were studied in terms of immunomodulatory activity. All showed the ability to stimulate the transformation of lymphocytes, and this time, the 96% ethanolic extract is considered the most active [48]. Furthermore, according to Shi and co-workers, the petroleum ether fraction of the product obtained from the 60% ethanol extraction of *Galium aparine* L. can exert anti-proliferative action in vitro on the leukemia cell line K562. This activity is due to the content rich in the three identified compounds: β-sitosterol, daucosterol, and dibutyl phthalate. The research team showed that the three compounds can inhibit the proliferation of the K562 leukemic cells in a concentration- and time-dependent manner. Butyl phthalate showed the strongest action, followed by β-sitosterol, these being considered the main contributors to the biological action of *G. aparine* L. extract [93].

The outcomes reported in the present study do not offer sufficient information regarding the effect of *Galium verum* L. extracts (ethanolic and ethyl acetate phases) on melanoma cells; therefore, future studies are needed regarding the antitumor potential of this plant material on human skin cancer, especially experiments that describe the molecular mechanism of the phenolic compounds, thus explaining the antitumor efficiency of ethanolic and ethyl acetate extracts of *Galium verum* L. In the current study, we have considered each *Galium verum* L. extract as an active ingredient because, in applied ethnomedicine, most of the biologically active compounds, present in a small amount, remain undetectable [94]. Both *Galium verum* L. extracts are a mixture of bioactive nutraceuticals, able to mediate therapeutic activity, but, with all these, their isolation should be performed to better explain the phytochemical basis of biological effects observed in the current study.

Nevertheless, summarizing the literature data and the results obtained in this study, *Galium verum* L. could be considered a promising nutraceutical for the antitumor approach to skin cancer.

## 4. Materials and Methods

### 4.1. Chemicals, Reagents, and Bacterial Strains

Ethanol 95% (*v*/*v*) purchased from Girelli alcool SRL (Milano MI, Italy) and ethyl acetate (≥99.5%), acquired from Sigma Aldrich (Steinheim, Germany) was used to obtain the extracts from the aerial parts of *Galium verum* L. plant material. To investigate the antioxidant potential through the DPPH method, 2,2-diphenyl-1-picrylhydrazyl (DPPH, purchased from Sigma Aldrich, Steinheim, Germany) was used. To compare the results of the extracts, ascorbic acid (acquired from Lach-Ner Company (Prague, Czech Republic)) was used as standard. All chemicals used were of high analytical-grade purity.

The standards used for the LC-MS analysis were as follows: chlorogenic acid, 4-O-caffeoylquinic acid, rutin, quercetin, quercetol, quercitrin, isoquercitrin were purchased from Sigma-Aldrich (St. Louis, MO, USA), while luteolin and gallic acid were purchased from Roth (Karlsruhe, Germany). The standards (+)-catechin, (−)-epicatechin, vanillic acid, syringic acid, and protocatechuic acid (3,4-dihydroxybenzoic acid) were purchased from Sigma-Aldrich (Steinheim, Germany), Merck (Darmstadt, Germany), and Alfa-Aesar (Karlsruhe, Germany). Methanol and acetic acid of HPLC analytical grade were purchased from Merck (Darmstadt, Germany). Ultrapure deionized water was provided by a MiliQ system Milli-Q^®^ Integral Water Purification System (Merck Millipore, Darmstadt, Germany).

The total phenolic content determination was performed using gallic acid 98% and Na_2_CO_3_ 99%, which were procured from Roth (Dautphetal, Germany), and Folin–Ciocalteu reagent, acquired from Merck (Darmstadt, Germany). The total flavonoid content determination was conducted using NaNO_2_ acquired from Merck, AlCl_3_ 98% purchased from Roth, and NaOH pellets procured from ChimReactiv SRL (Bucharest, Romania). The standard used for the determination of flavonoid content was (+)-Catechin hydrate 98%, acquired from Sigma-Aldrich.

For the in vitro experiments, the reagents used were culture medium–high glucose Dulbecco’s Modified Eagle’s Medium (DMEM) and the cell culture supplement fetal bovine serum (FBS) and trypsin-EDTA solution were purchased from PAN-Biotech GmbH (Aidenbach, Germany). Penicillin/streptomycin (Pen/Strep-10,000 IU/mL), phosphate saline buffer (PBS), dimethyl sulfoxide (DMSO-solvent), and MTT (3-(4,5-dimethylthiazol2-yl)-2,5-diphenyltetrazolium bromide) viability kit were procured from Sigma-Aldrich, Merck KgaA (Darmstadt, Germany).

For the antibacterial potential assaying of the *Galium verum* extracts, all microorganism strains were acquired from the American Type Culture Collection (ATCC) (Manassas, VA, USA). The following aerobic bacterial strains, representative of the human pathogenic bacteria, were used: two Gram-positive *Staphylococcus aureus* (ATCC 25923) and *Streptococcus pyogenes* (ATCC 19615) and two Gram-negative *Escherichia coli* (ATCC 25922) and *Pseudomonas aeruginosa* (ATCC 27853). Initially, all tested bacteria were isolated on Columbia agar with 5% sheep blood (ThermoScientific, Waltham, MA, USA).

### 4.2. Cell Culture

In vitro experiments were realized on HaCaT-human keratinocytes (CLS, CVCL_0038) provided by the Cell Lines Service GmbH (Eppelheim, Germany) and on A375-human melanoma cell line (ATCC^®^ CRL-1619™) purchased from the American Type Culture Collection (ATCC, Manassas, VA, USA). HaCaT and A375 cells were cultured and grown in DMEM supplemented with 10% FBS and 1% antibiotic mixture (Pen/Strep). The analyses were performed under standard conditions: a humidified atmosphere with 5% CO_2_ and 37 °C. Cells were stimulated with Galium verum extracts at various concentrations (15–55 μg/mL).

### 4.3. Plant Material and Extraction Technique

The plant material (dried aerial parts—herba) of the *Galium verum* L. species was purchased from the AdNatura store (S.C. ADSERV S.R.L, Timisoara, Romania, batch no. 11/2022) and kept at room temperature (22 ± 2 °C), and then crushed before being subjected to extraction using the following types of solvents: ethanol 95% and a mixture of distilled water and ethyl acetate (≥99.5%), according to the procedure described in the literature, slightly modified [80].

The extraction procedure was carried out as follows: initially, 25 g of dried and ground plant product was mixed with 150 mL of 95% ethanol and covered with parafilm, and the whole mixture was left to maceration for 24 h. After 24 h at room temperature (22 ± 2 °C), the extract was subjected to ultrasound for 30 min, using an Elma S120 Elmasonic ultrasonic water bath, then filtered through a Whatman grade 4 filter paper, followed by another filtration through a 0.45 μm nylon membrane filter (Agilent Technologies, Santa Clara, CA, USA), to ensure the sterilization of the extract. To remove the solvent (ethanol), a rotary vacuum evaporator (HEIDOLPH Laborata 4000 efficient WB eco) was used at a temperature of 25 °C and a pressure of 60 mbar. The extract obtained (hereafter referred to as GvEtOH) was stored in a refrigerator at 4 °C until further evaluation and use [95]. Furthermore, the 25 g of plant material residue initially used in the first phase of extraction was weighed at 10 g, over which a mixture of 150 mL of distilled water and 200 mL of ethyl acetate was added, and the Erlenmeyer flask was sealed with parafilm. After another 24 h of maceration, the mixture was sonicated for 30 min, followed by a separation of the two phases (first, the ethyl acetate phase, followed by the aqueous phase). The ethyl acetate phase was collected, and the extract was concentrated using a rotary evaporator at a temperature of 25 °C (to avoid possible degradation of vegetal product) and a pressure of 130 mbar. The obtained extract (GvEtOAc) was kept under the same conditions as the initial total alcoholic extract (GvEtOH) [96].

For biological experiments, each concentrated extract was diluted with 0.5% DMSO to yield a final stock solution of 1 mg/mL. For phytochemical investigation, the extracts were diluted in 95% ethanol to yield the same final stock solution of 1 mg/mL. The stock solutions obtained were kept at 4 °C until further evaluation. The schematic protocol is depicted in Figure 10.

After processing the dried plant material and using the conventional methods of extraction (maceration followed by sonication), we can determine the extraction efficiency, namely the extraction yield, taking into account the parameters that were used in the working protocol: extraction time, plant/solvent ratio, contact time between plant and solvent and solvent concentration. For the extraction yield calculation, 50 mL of each extract obtained (ethanol and ethyl acetate) was subjected to an evaporator at a constant temperature of up to 25 °C to avoid phytocompounds degradation. The total volume of each extract obtained after the extraction procedure was 118 mL in the case of GvEtOH phase and 156 mL for GvEtOAc phase. The extraction yield was calculated using the following equation:(1)$$\eta \;[\%]\;=\frac{m_{residue}\;\cdot \;V_{extract\;}}{V_{total}\;\cdot \;m_{plant\;\;material}}\;\cdot \;100$$
where: *η*—extraction yield (%); $m_{residue}$—the mass of the residue obtained after concentration (g); $V_{extract}$—the volume of the vegetal extract samples subjected to concentration step (mL); $V_{total}$—the total volume of vegetal extract samples obtained after the extraction process (mL); $m_{plant\;material}$—the amount of the plant material used in the extraction process (g).

### 4.4. Phytochemical Screening

#### 4.4.1. FT-IR

To identify the presence of chemical molecules in both *Galium verum* L. extracts, the Fourier transform infrared spectroscopy (FT-IR) was employed using a Prestige-21 spectrometer (Shimadzu, Duisburg, Germany). The work conditions were as follows: room temperature (22 ± 2 °C), a spectral region ranging from 4000 to 400 cm^−1^ using KBr pellets, and a resolution of 4 cm^−1^. The FT-IR investigation is a qualitative method that allows us to identify the functional organic groups of the main polyphenols present in the dried *Galium verum* extracts based on the perfect match between the recorded absorption bands of the dried extracts at a specific wavenumber and the absorption bands frequencies from the library [97].

#### 4.4.2. Liquid Chromatography Mass Spectrometry (LC-MS) Analysis

To identify the polyphenols that are present in both extracts, liquid chromatography coupled with mass spectrometry (LC/MS) was performed using the Agilent Technologies 1100 HPLC Series system (Agilent, Santa Clara, CA, USA), according to a previously validated and described method [81,98,99]. The system was equipped with a degasser (G1322A), binary gradient pump (G13311A), column thermostat, auto-sampler (G1313A), and UV detector (G1316A). In addition, the HPLC system was also coupled with an Agilent 1100 mass spectrometer (LC/MSD Ion Trap SL). A reverse-phase analytical column (Zorbax SB-C18 100 × 3.0 mm i.d., 3.5 μm particle) was employed for the separation at a work temperature of 48 °C. The detection of the compounds present in both extracts was performed on both UV and MS modes. For the analysis, the UV detector was set at a wavelength of 330 nm for 17 min. (to detect the polyphenolic acids), and then at 370 nm wavelength for 38 min. (to detect flavonoids and their aglycones). By using an electrospray ion source in negative mode (capillary +3000 V, nebulizer 60 psi (nitrogen), dry gas nitrogen at 12 L/min. and dry gas temperature 360 °C), the MS system was put into operation [58]. To carry out the analysis, a mobile phase, which consists of a binary gradient formed from methanol and acetic acid 0.1% (*v*/*v*), was used. The first elution (5% methanol), which lasted for 35 min with a flow rate of 1 mL·min^−1^ for 5 μL injection volume, started with a binary linear gradient and ended at 42% methanol; then, the isocratic elution started with 42% methanol and lasted 3 min, followed by the rebalancing of the column with 5% methanol, which lasted for 7 min [100]. To identify the compounds, the MS spectra obtained from the standard solution of polyphenols were inserted in a mass spectra library and compared with the MS spectra/traces of each polyphenol found in the test solutions, provided by an MS signal used only for qualitative analysis. Based on the standard-compound spectral match, the polyphenols present in both *Galium verum* L. extracts were identified. From MS detection, the UV trace of identified compounds was used for their quantification. In the case of compounds whose MS spectra overlap, they can be selectively identified based on the differences between their molecular mass and the MS spectra obtained from qualitative analysis (MS detection). The quantification and detection limits for each compound were 0.1 μg/mL. The detection limits were calculated as minimal concentration, producing a reproductive peak with a signal-to-noise ratio greater than three. By using an external standard method, the quantitative determinations were performed. By using ChemStation (vA09.03) and Data Analysis (v5.3) software from Agilent (Santa Clara, CA, USA), all the chromatographic data were processed [100]. The calibration curves of their corresponding standards for a five-point plot in the range of 0.1–50 μg/mL, with suitable linearity (R^2^ = 0.999), were used to determine the concentration of polyphenols in *Galium verum* L. extract samples, and the results were expressed as μg of polyphenolic compound/mL of *Galium verum* L. extract.

Using the same analytical conditions described above, catechin, epicatechin, gallic acid, syringic acid, vanillic acid, and protocatechuic acid were investigated. The only applied difference was the elution, which started with a different binary gradient and compound detection in MS mode. The binary gradient started with 3% methanol and lasted over 3 min., followed by 8% methanol for 8.5 min., then 20% methanol for 10 min., and finally 3% methanol to rebalance column. The investigated compounds were quantified based on their peak area and the calibration curve of their corresponding standards, and the results were expressed as μg of polyphenolic compound/mL of *Galium verum* L. extract [58].

#### 4.4.3. Total Phenolic (TPC) and Flavonoid Contents (TFC)

The total phenolic content of both extracts obtained from the *Galium verum* L. aerial part was performed using the Folin–Ciocalteu method, with some modifications [77]. The method is based on mixing 0.5 mL *Galium verum* L. extract solution (1 mg/mL) with 2.5 mL Folin–Ciocalteu reagent, which was previously diluted 1:10. Then, 2 mL of 7.5% Na_2_CO_3_ solution was added in each mixture. Both samples were kept in the dark for 90 min, and then the absorbance was read versus blank at 750 nm wavelength using a UviLine 9400 Spectrophotometer from SI Analytics (Mainz, Germany). For the quantification of the total phenolic content, an equation obtained from the calibration curve of gallic acid was used (R^2^ = 0.997), using gallic acid solutions of different concentrations (0.05–1 mg/mL). The total phenolic content of both *Galium verum* L. extracts was expressed as milligrams of gallic acid equivalents (GAE) per gram of dry extract (mg GAE/g dry extract).

The total flavonoid content of both extracts obtained from *Galium verum* L. herba was conducted using the method described by Masaada and co-workers [101]. Briefly, 250 μL of 1 mg/mL from each extract solution was mixed with 75 μL of 5% NaNO_2_ solution. After 6 min, the mixture was added sequentially 150 μL of 10% AlCl_3_ and 500 μL of 1 M NaOH. After that, the total volume was adjusted to 2.5 mL by completing with distilled water. The absorbance was read at 510 nm wavelength versus blank using a UviLine 9400 Spectrophotometer from SI Analytics. The total flavonoid content was calculated using an equation obtained from the calibration curve of (+)-Catechin hydrate (R^2^ = 0.999) in the range of 0.001–0.05 mg/mL. The results were expressed as milligrams of catechin equivalents per gram of dry extract (mg CE/g dry extract).

#### 4.4.4. Antioxidant Activity

The antioxidant potential (AP) of *Galium verum* ethanol and ethyl acetate extracts (at 6 different concentrations for EC_50_ determination) was established using a DPPH free radical-scavenging assay, according to a previously reported method, modified and developed by our research group [102]. The results obtained were expressed as EC_50_ value, which represents the half maximal inhibitory concentration of the antioxidants contained in *Galium verum* L. total ethanolic extract as well as in the *Galium verum* L. ethyl acetate fraction, needed to scavenge 50% of DPPH free radicals present in the test solutions. Briefly, an ethanol solution of DPPH 0.1 mM was prepared and kept at 4 °C until further use. A precise volume of each test sample was added into a quartz test cuvette (10 × 10 mm) with 2.7 mL DPPH 0.1 mM ethanol solution. The absorbance values were read continuously for 20 min using a UviLine 9400 Spectrophotometer from SI Analytics (Mainz, Germany). As etalon for comparison, it was used ascorbic acid (Vit C) 0.4 mg/mL in 95% ethanol. The absorbances (test samples, etalon, and control) were measured spectrophotometrically at 517 nm wavelength. For the quantification of the DPPH free radical inhibition percentage, the below equation was used:(2)$$AP\;(\%)\;=\;(\frac{A_{DPPH}\;-\;A_{test\;sample}}{A_{DPPH}})\;\cdot \;100$$
in which the $A_{test\;sample}$ is the absorbance of each concentration of *Galium verum* L. test sample (from total ethanolic extract and ethyl acetate fraction) in the presence of DPPH free radical and $A_{DPPH}$ is the absorbance of DPPH free radical (control) without the *Galium verum* L. test sample.

By linear regression analysis curve plotting between the inhibition percentages of antioxidant potential (AP%) obtained and the concentrations of each test sample of *Galium verum* L. (total ethanolic extract and ethyl acetate fraction), the half maximal inhibitory concentration (EC_50_) was determined using OriginLab 2020b software (Origin Lab—Data Analysis and Graphing Software, Szeged, Hungary).

### 4.5. Bioactivity Screening

#### 4.5.1. In Vitro Antimicrobial Effects

Antimicrobial activities of the *Galium verum* L. extracts were tested by determination of minimum inhibitory concentration (MIC) and minimum bactericidal concentration (MBC). The broth dilution assay was performed according to indications from both the European Committee on Antimicrobial Susceptibility Testing (EUCAST) and the Clinical Laboratory and Standard Institute (CLSI) and was extensively described in previous studies [103,104,105,106,107]. The standardized bacterial inoculum of 0.5 McFarland was diluted in NaCl 0.85% (bioMérieux, Marcy-l’Étoile, France) to obtain approximately 5 × 10^5^ colony-forming units/mL (CFU). Then, the bacterial suspension and the tested compounds were added in Mueller Hinton broth (ThermoScientific, Waltham, MA, USA), supplemented with blood and β-Nicotinamide adenine dinucleotide (β-NAD) for *S. pyogenes*, obtaining dilutions with concentrations of 30, 15, 7.5 and 3.75 mg/mL. After 24 h incubation at 35 °C, the lowest concentration without visible growth was interpreted as the MIC value. The MBC was established by sub-cultivating on Columbia agar with 5% sheep blood 1 µL of suspension from the test tube without visible growth. The lowest concentration that killed 99.9% of the bacteria was considered as MBC. Determinations were performed in triplicate for each tested strain and each *Galium verum* L. extract.

#### 4.5.2. Anticancer Potential

##### Cell Viability Assessment

*G. verum* L. extracts were evaluated for possible anticancer activity in vitro against the human skin cancer cell line A375, and in addition, their effect on the non-tumor skin cell line (HaCaT) was evaluated.

The MTT (3-(4,5-dimethylthiazol-2-yl)-2,5-diphenyltetrazolium bromide) colorimetric assay was applied to evaluate the impact of *G. verum* extracts on cell viability. Briefly, cancerous and non-cancerous cells were seeded (1 × 10^4^ cells/well) in 96-well culture plates and allowed to adhere to the bottom of the well. After they attached, the cells were stimulated with different concentrations of the two extracts dissolved in DMSO (15, 25, 35, 45, and 55 μg/mL) and incubated for 24 h. The control group was represented by untreated cells. After the incubation period, the culture medium was exchanged with 100 μL/well of fresh medium, and then the cells were treated with 10 μL/well of MTT reagent 1 solution (tetrazolium salt) for 3 h. In the end, the blue formazan crystals obtained were dissolved in 100 μL of solution 2 (solubilization buffer) from the MTT kit and left in contact for 30 min, as presented in one of our previous publications [108]. Absorbance was determined at 570 nm with a Cytation 5 device (BioTek Instruments Inc., Winooski, VT, USA) to calculate cell viability. All experiments were carried out in triplicate.

##### Cell Morphology and Confluence Evaluation

To comprehend the cytotoxic potential effect induced in A375 and HaCaT cells, cellular morphology and confluence were analyzed after 24 h of stimulation with *G. verum* L. extracts. Cells were microscopically examined under bright field illumination. The images were taken using Cytation 1 (BioTek Instruments Inc., Winooski, VT, USA) and interpreted using Gen5 Microplate Data Collection and Analysis Software (BioTek Instruments Inc., Winooski, VT, USA).

##### Nuclear Staining Evaluation

To determine the type of cell death (apoptosis/necrosis) induced by the analyzed *G. verum* L. extracts, the Hoechst 33342 test was performed. Briefly, cells were cultured in 12-well plates (1 × 10^5^ cells/well) and then, at an appropriate level of confluence, were stimulated with increasing concentrations (15 and 55 μg/mL). After 24 h, the cell medium was removed, and 500 μL/well of staining solution diluted 1:2000 in PBS was added. After keeping it at room temperature in the dark for 10 min, the solution was removed and washed three times with PBS. The pictures were captured using Cytation 1 and processed using Gen5 Microplate Data Collection and Analysis Software. The apoptotic index was calculated according to the formula reported by Xu and co-workers [109].

### 4.6. Statistical Analysis

The results are expressed as mean ± standard deviation (SD). GraphPad Prism software version 9.4.0 for Windows (GraphPad Software, San Diego, CA, USA, www.graphpad.com) was used to present statistical data obtained in all the in vitro biological studies. A one-way ANOVA test followed by Dunnett’s multiple post-test comparisons was utilized to determine statistical differences in samples. Statistically significant differences between data were tagged with * *p* < 0.1, ** *p* < 0.01, *** *p* < 0.001, and **** *p* < 0.0001. OriginLab 2020b software (Origin Lab—Data Analysis and Graphing Software, Szeged, Hungary) was used to process the statistical data obtained at the antioxidant potential and FT-IR investigations of *Galium verum* L. extracts.

## 5. Conclusions

The present study undertakes an evaluation of the phytochemical and biological profile of two phases of *Galium verum* L. plant material, especially their efficacy on skin cancer. In the present study, two extraction phases (ethanol and ethyl acetate) of Galium verum L. were prepared and assessed, as well as phytocompound, antimicrobial, and antitumor properties. The antioxidant screening outcomes showed that, following the evaluation of several concentrations of *Galium verum* L. ethanolic extract and ethyl acetate fraction samples, all the samples tested have significant antioxidant potential in a concentration dose-dependent manner and an EC_50_ value quite suitable as compared with polyphenols content determined through LC-MS analysis. GvEtOAc extract showed an EC_50_ value much higher than GvEtOH extract, meaning that the ethyl acetate phase contains a higher amount of antioxidants than the ethanolic one. Rutin, chlorogenic acid, and isoquercitrin were the phenolic compounds found in high concentrations in both extracts, and we believe that they are responsible for the investigated activities in this study. The ethyl acetate phase recorded more concentrated phenolic compounds than ethanolic one, a result also confirmed by total phenolic content determination (1.39 mg GAE/g dry extract for GvEtOAc vs. 1.30 mg GAE/g dry extract for GvEtOH). However, the total quantity of flavonoids resulting from LC-MS analysis was higher for GvEtOH extract than GvEtOAc extract, a fact also confirmed by the total flavonoid content determination (1.42 mg CE/g dry extract for GvEtOH vs. 1.37 mg CE/g dry extract for GvEtOAc).

The antimicrobial activity outcomes confirm one more evidence of the effectiveness of the traditional use of *Galium verum* L. herba against various pathogens, especially against the Gram-positive *Streptococcus pyogenes* and *Staphylococcus aureus* bacilli strains, the ethyl acetate phase being more active then ethanolic one. One can affirm that the changes observed in the antimicrobial activity of *Galium verum* L. extracts corresponded to the type of solvent used. Regarding the in vitro antitumor tests performed, the outcomes suggested that the *Galium verum* L. extracts showed a potential dose-dependent cytotoxic effect against A375 melanoma cell lines. The more pronounced activity is again revealed by the ethyl acetate phase (GvEtOAc).

In summary, we can conclude that our results complete the lack of literature data with new information concerning the bioactivity of *Galium verum* L. herba natural product, especially regarding the antitumor potential on malignant melanoma cells.

## Figures and Tables

**Figure 1 molecules-28-07804-f001:**
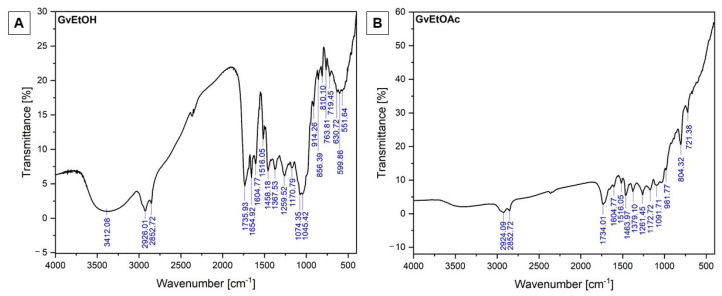
FT−IR spectra of ethanol (**A**) and ethyl acetate (**B**) *Galium verum* L. extracts.

**Figure 2 molecules-28-07804-f002:**
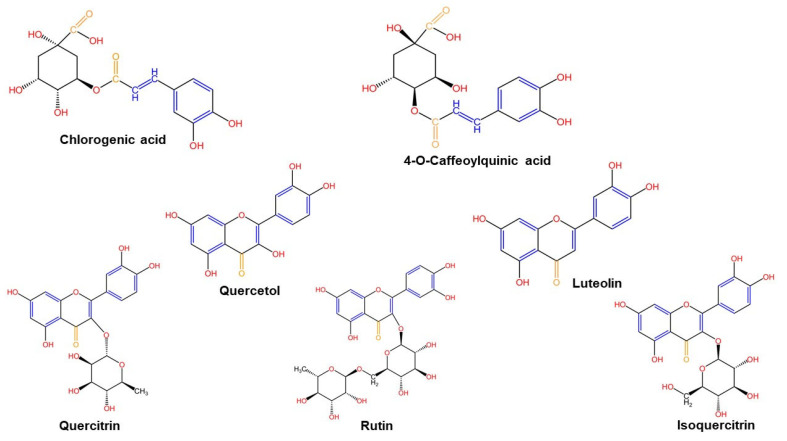
Chemical structures of the main polyphenols found in both *Galium verum* L. extracts after LC-MS analysis.

**Figure 3 molecules-28-07804-f003:**
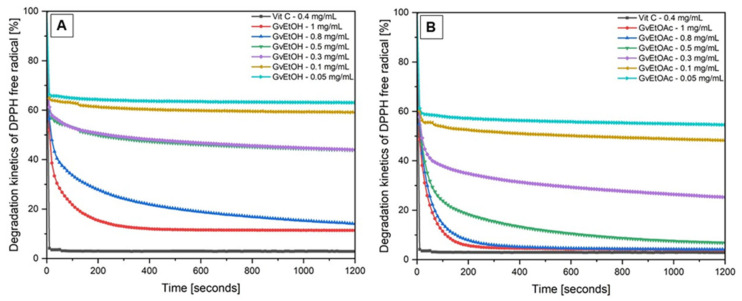
The time-dependent degradation kinetics of DPPH free radicals are provided by the ethanol (**A**) and ethyl acetate (**B**) *Galium verum* L. extracts as well as by the ethanolic solution of vitamin C (black line).

**Figure 4 molecules-28-07804-f004:**
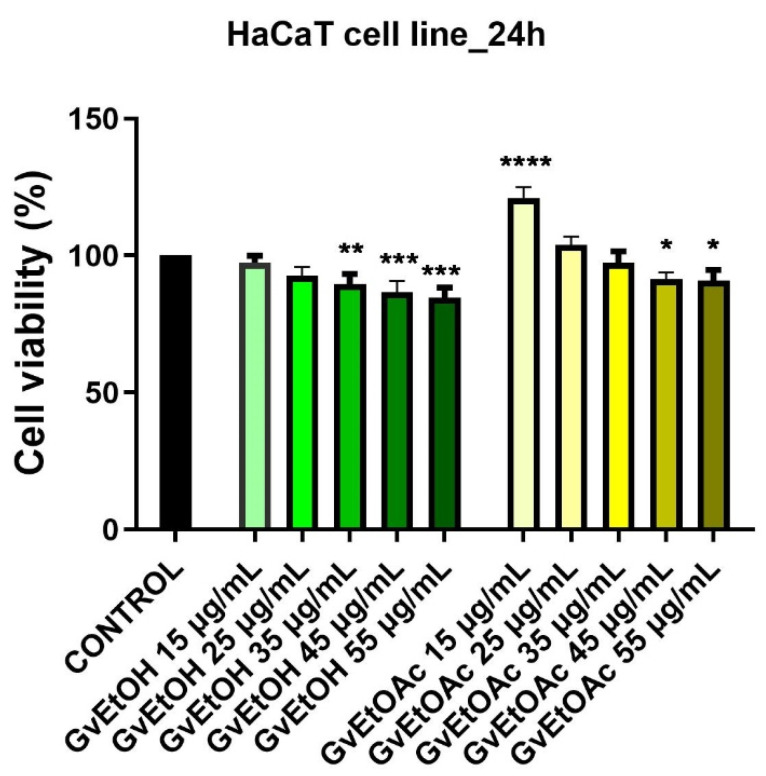
Cell viability effect of GvEtOH and GvEtOAc extracts (15, 25, 35, 45, and 55 μg/mL) determined by the MTT assay, 24 h post-stimulation of HaCaT immortalized human keratinocytes. The statistical differences between the control and the treated group were analyzed by applying the one-way ANOVA analysis followed by Dunett’s multiple comparisons post-test (* *p* < 0.05; ** *p* < 0.01; *** *p* < 0.001; **** *p* < 0.0001).

**Figure 5 molecules-28-07804-f005:**
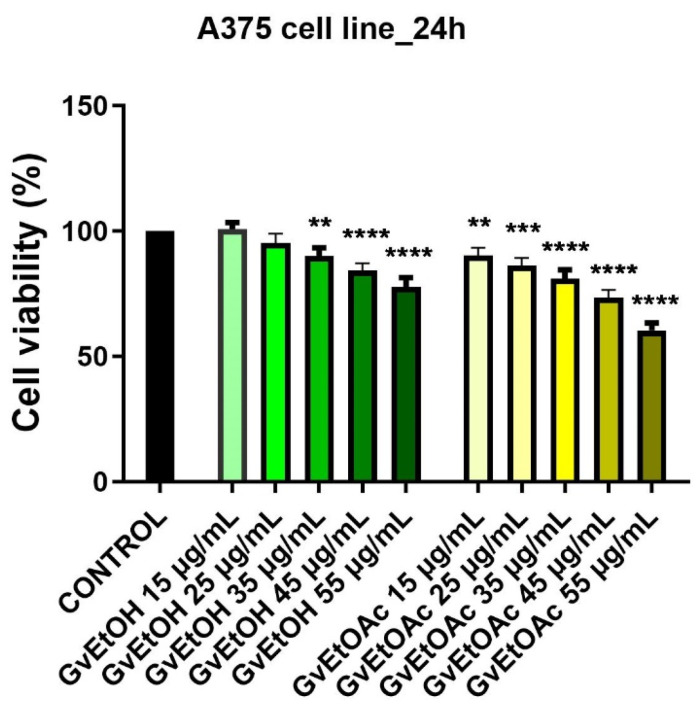
Cell viability effect of GvEtOH and GvEtOAc extracts (15, 25, 35, 45, and 55 μg/mL) determined by the MTT assay, 24 h post-stimulation of A375 human melanoma cells. The statistical differences between the control and the treated group were analyzed by applying the one-way ANOVA analysis followed by Dunett’s multiple comparisons post-test (** *p* < 0.01; *** *p* < 0.001; **** *p* < 0.0001).

**Figure 6 molecules-28-07804-f006:**
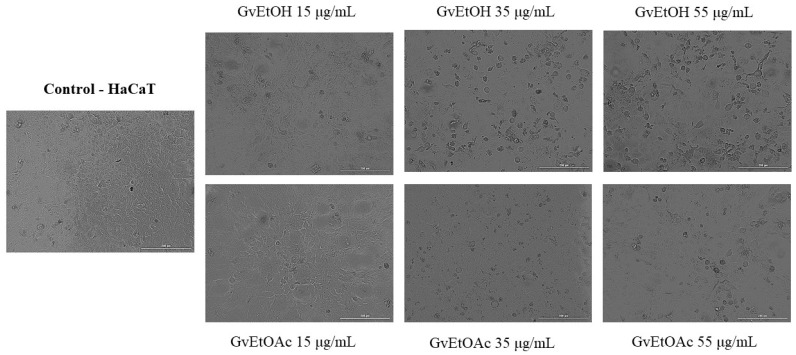
Morphology and confluence of HaCaT cells following the 24 h of treatment with GvEtOH and GvEtOAc (15, 35, and 55 µg/mL). The scale bars indicate 200 µm.

**Figure 7 molecules-28-07804-f007:**
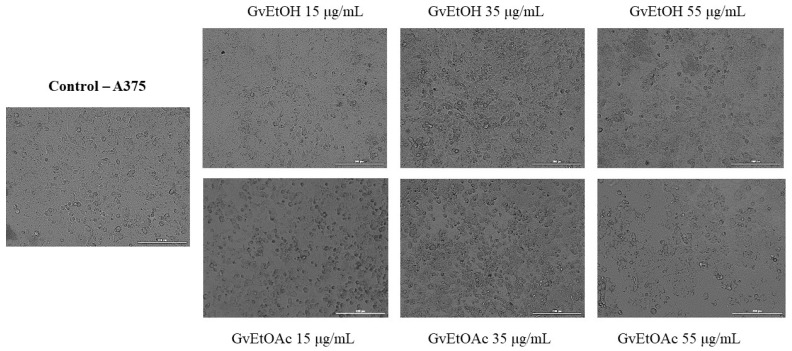
Morphology and confluence of A375 cells following the 24 h of treatment with GvEtOH and GvEtOAc (15, 35, and 55 µg/mL). The scale bars indicate 200 µm.

**Figure 8 molecules-28-07804-f008:**
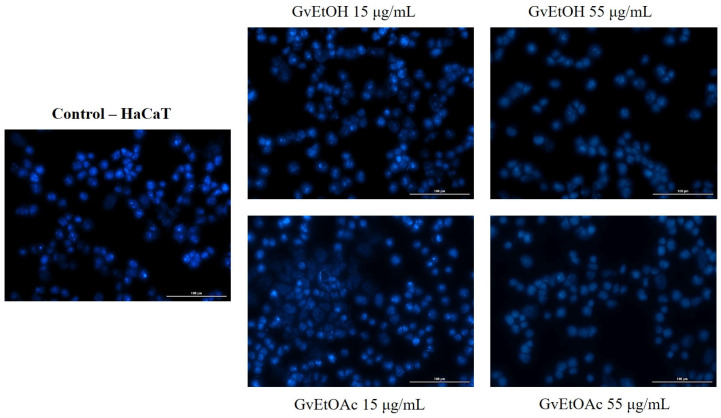
HaCaT nuclei stained with Hoechst 33342 dye after 24 h of treatment with GvEtOH and GvEtOAc (15 and 55 µg/mL). The scale bars represent 100 µm.

**Figure 9 molecules-28-07804-f009:**
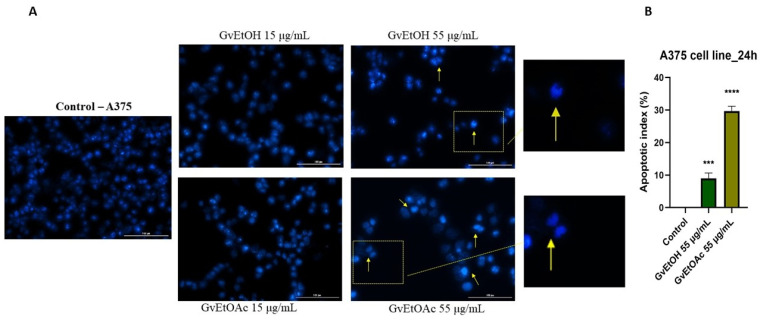
(**A**) A375 nuclei stained with Hoechst 33342 dye after 24 h of treatment with GvEtOH and GvEtOAc (15 and 55 µg/mL) and (**B**) calculated apoptotic index (AI) percentages for the highest concentration tested (75 µg/mL). The yellow arrows indicate signs of apoptosis. The scale bars represent 100 µm. Data are presented as an apoptotic index (%) normalized to control and expressed as mean values ± SD of three independent experiments. The statistical differences between the control and the treated group were analyzed by applying the one-way ANOVA analysis followed by Dunett’s multiple comparisons post-test (*** *p* < 0.001; **** *p* < 0.0001).

**Figure 10 molecules-28-07804-f010:**
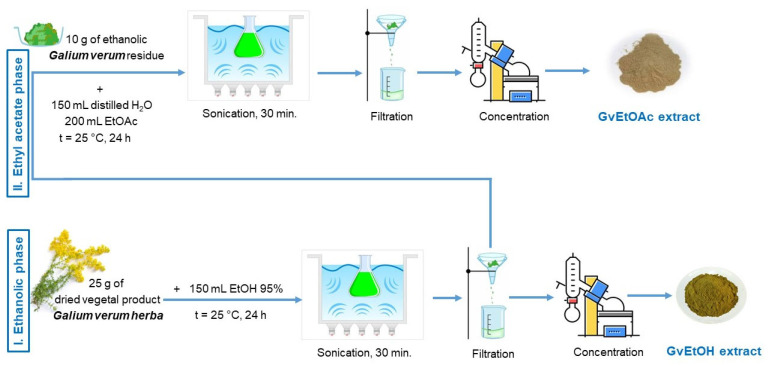
Schematic protocol of *Galium verum* L. extracts preparation.

**Table 1 molecules-28-07804-t001:** Peak values and functional groups of ethanol and ethyl acetate *Galium verum* L. extracts recorded in the spectrum.

Wavenumber (cm^−1^)		Functional Groups		Bond	
GvEtOH	GvEtOAc	GvEtOH	GvEtOAc	GvEtOH	GvEtOAc
3412.08	-	Alcohol	-	OH stretch (H-bonded)	-
2926.01	2924.09	Alkane/ Alcohol (acid)	Alkane/ Alcohol (acid)	C-H stretching/ OH stretch	C-H stretching/ OH stretch
2852.72	2852.72	Alkane	Alkane	C-H stretching	C-H stretching
1735.93	1734.01	Carbonyl	Carbonyl	C=O stretch	C=O stretch
1654.92	-	Amide/Alkene	-	C=O stretch C=C stretching	-
1604.77	1604.77	Cyclic alkene/	Cyclic alkene/	C=C stretching	C=C stretching
1516.05	1516.05	Aromatic compounds	Aromatic compounds	C=C stretch	C=C stretch
1458.18	1463.97	Alkane (methylene group)/Aromatics	Alkane (methylene group)/Aromatics	C-H bending/ C=C stretch (in ring)	C-H bending/C=C stretch (in ring)
1367.53	1379.10	Alkane	Alkane	-C-H bending	-C-H bending
1259.52	1261.45	Acids	Acids	C-O stretch	C-O stretch
1170.79	1172.72	Alcohol (tertiary)	Alcohol (tertiary)	C-O stretching	C-O stretching
1074.35	1091.71	Alcohol (primary)	Alcohol (secondary)	C-O stretch	C-O stretch
1045.42	-	Anhydride	-	CO-O-CO stretching	-
914.26	981.77	Alkane (disubstituted (trans))/Alkenes	Alkane (disubstituted (trans))/Alkenes	C=C bending/ =C-H bending	C=C bending/ =C-H bending
856.39	-	Alkane	-	C=C bending	-
810.10	804.32	Alkane (trisubstituted)/ Halo compounds	Alkane (trisubstituted)/ Halo compounds	C=C bending/ C-Cl stretching	C=C bending/ C-Cl stretching
763.81	-	Alkane (trisubstituted)/ Halo compounds	-	C=C bending/ C-Cl stretching	-
719.45	721.38	Alkane (disubstituted (cis))/ Halo compounds	Alkane (disubstituted (cis))/ Halo compounds	C=C bending/ C-Cl stretching	C=C bending/ C-Cl stretching
630.72	-	Halo compounds	-	C-Cl; C-Br stretching	-
599.86	-	Halo compounds	-	C-Cl; C-Br stretching	-
551.64	-	Halo compounds	-	C-Cl; C-Br stretching	-

**Table 2 molecules-28-07804-t002:** Polyphenolic compounds of both extracts analyzed by LC-MS.

GvEtOH			
Compound Name	UV Identified	MS Qualitatively Identified	Concentration (μg/mL)
Chlorogenic acid	Yes	Yes	8.027
4-O caffeoylquinic acid	Yes	Yes	0.172
Isoquercitrin	Yes	Yes	17.765
Rutin	Yes	Yes	14.811
Quercitrin	No	Yes	-
Quercetol	Yes	Yes	1.275
Luteolin	Yes	Yes	0.260
GvEtOAc			
Chlorogenic acid	Yes	Yes	10.216
4-O caffeoylquinic acid	Yes	Yes	0.096
Isoquercitrin	Yes	Yes	20.384
Rutin	Yes	Yes	1.896
Quercitrin	Yes	Yes	6.722
Quercetol	Yes	Yes	0.779
Luteolin	Yes	Yes	0.191

**Table 3 molecules-28-07804-t003:** Catechins content of *Galium verum* L. extracts by LC-MS.

Extract	Concentrations (μg/mL)					
	Epicatechin	Catechin	Syringic Acid	Gallic Acid	Protocatechuic Acid	Vanillic Acid
GvEtOH	1.10	ND ^1^	ND ^1^	ND ^1^	ND ^1^	ND ^1^
GvEtOAc	ND ^1^	ND ^1^	ND ^1^	0.34	ND ^1^	ND ^1^

^1^ ND—not detected

**Table 4 molecules-28-07804-t004:** The antioxidant potential values (%) of *Galium verum* extracts at six concentrations tested as compared with vitamin C (standard) and the corresponding EC_50_ values.

Examined Extract Concentration (mg/mL)	Antioxidant Potential AP (%)			EC_50_ (mg/mL)	
	Vitamin C (Standard)	GvEtOH	GvEtOAc	GvEtOH	GvEtOAc
1	97.08 ± 0.04	88.66 ± 0.003	96.10 ± 0.04	0.136 ± 0.03	0.074 ± 0.01
0.8		85.95 ± 0.06	95.91 ± 0.04		
0.5		56.21 ± 0.04	93.24 ± 0.04		
0.3		56.05 ± 0.04	74.74 ± 0.06		
0.1		40.87 ± 0.07	51.74 ± 0.04		
0.05		36.96 ± 0.02	45.41 ± 0.03		

**Table 5 molecules-28-07804-t005:** The minimum inhibitory concentration (MIC) and the minimum bactericidal concentration (MBC) values.

Test Compounds	Microbial Strains	MIC (mg/mL)	MBC (mg/mL)
GvEtOH	*Streptococcus pyogenes (Gram +)*	15	30
	*Staphylococcus aureus (Gram +)*	30	30
	*Escherichia coli (Gram −)*	30	NA ^1^
	*Pseudomonas aeruginosa (Gram −)*	NA ^1^	NA ^1^
GvEtOAc	*Streptococcus pyogenes(Gram +)*	15	15
	*Staphylococcus aureus (Gram +)*	15	15
	*Escherichia coli (Gram −)*	30	NA ^1^
	*Pseudomonas aeruginosa (Gram −)*	NA ^1^	NA ^1^

^1^ NA—no activity (absent antimicrobial activity)

## Data Availability

Authors can provide raw data upon request.

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
