# Peer review of "Phytochemical and Nutraceutical Screening of Ethanol and Ethyl Acetate Phases of Romanian Galium verum Herba (Rubiaceae)"

_molecules, 2023, doi:10.3390/molecules28237804_

Round 1
Reviewer 1 Report
Comments and Suggestions for Authors
Authors have reported the Phytochemical and Nutraceutical Screening of ethanol and ethyl acetate phases of Romanian Galium verum herba (Rubiaceae) but the following need to be considered
1. References is needed for ''Nowadays, more and more emphasis is placed on the production of herbal medicines for the treatment of human diseases. Authors may go through https://doi.org/10.1016/j.sajb.2023.08.046 ; https://doi.org/10.1016/j.hermed.2011.11.001
2.rationalae of the studies must be given
3, Chemical structure of the major compounds must be given.
4. reference of antimicrobial is needed.
5. comparison with the standard drug need to be given for Antimicrobial, . Cell viability.
6. Check for an spelling errors
7. References need to be re-checked and should be in journals format.
Author Response
Please find the document attached.

Reviewer 2 Report
Comments and Suggestions for Authors
· The study did not provide sufficient information regarding the effect of Galium verum L. extracts on melanoma cells, indicating the need for future studies on the antitumor potential of this plant material on human skin cancer.
· The efficacy of Galium verum L. in skin cancer has not yet been established, and further research is needed to determine its effectiveness in this specific type of cancer.
· The study focused on the phytochemical profile and biological evaluation of the extracts, but did not investigate the mechanism of action or the specific compounds responsible for the observed effects.
· The study used in vitro cell lines for the evaluation of the anticancer effect, which may not fully represent the complexity of the in vivo tumor microenvironment.
I think the author could benefit from the below points could improve this work
· Provide a clear and detailed description of the extraction procedure, including the specific solvents used and their concentrations.
· Include information on the source and storage conditions of the plant material, such as the batch number and room temperature storage.
· Specify the duration of the stimulation period for the cells, such as the 24-hour time frame mentioned in the study.
· Clearly outline the concentrations of the extracts tested, including the lowest and highest concentrations used in the experiments.
· Include information on the methods used to evaluate cell viability, such as microscopic examination or other assays.
· Provide references to the literature for the extraction procedure, ensuring that any modifications made to the protocol are clearly stated.
Author Response
Please find the document attached.

Reviewer 3 Report
Comments and Suggestions for Authors
An interesting study about the screening of phytochemical and bioactivity analysis of Romanian galium verum herba was performed. The authors have shown the anti-oxidant, anti-microbial, and anti-cancer activity of the extract. Here are several comments that might improve the manuscript:
- Please proofread the manuscript for grammatical errors.
- Line 141, Please specify how the concentration of the extract was performed.
- Have you considered performing the total phenolic content of the extracts? it would be interesting to see how the phenolic compounds might be correlated with the anti-oxidant, antimicrobial, and anti-cancer activity of the extracts.
- In line 196, you mentioned that the flavonoids are higher than phenolic acids. could you please cite in the text the actual data (Figure/Table) that you are referring to.
- Line 279 - You mentioned that the GcEtOAc extract was able to stimulate the proliferation of healthy cell lines, however, this was only observed in lower doses (we would expect this effect would be greater in greater doses however no dose-response trend was seen) which was also confirmed by lower confluency in higher doses (Figure 5). Please justify how this could happen?
-Figure 5 and 6, if possible, could you please show an image with a bright/white background? Please also use a figure with high resolution. as at the moment in the current figure, it is hard to see the cells.
- Figures 7 and 8, please use images with high resolution.
- Line 251 - Anti-microbial analysis. Why was this performed? how does the data fit in your main aim which was focusing on the anti-cancer activity of the extract? What was the basis for choosing these bacteria for this study? Please explain in the discussion.
- Table 2. Have you run any statistical analysis to compare the compound concentration between both extracts? Are they the same and not significantly different?
- Line 874 - You mentioned that rutin, chlorogenic acid, and isoquercitrin were responsible for the investigated activities in this study. Do you have any data or correlation tests to prove this argument? How does the phytochemical composition of the extract affect the extract activity? here we can see that Rutin is much higher in GvETOH compared to GvETOAc. However, GvETOAc seems to have a better anti-oxidant activity.
- The title mentioned "nutraceutical" - however throughout the main text of the article, nothing related to nutraceuticals was mentioned. How are the findings in this study related to nutraceuticals?
Comments on the Quality of English Language
- Please proofread the manuscript for grammatical errors.
Author Response
Please find the document attached.

Round 2
Reviewer 1 Report
Comments and Suggestions for Authors
comments incorporated